# A hexa-species transcriptome atlas of mammalian embryogenesis delineates metabolic regulation across three different implantation modes

Anna Malkowska 1,2, Christopher Penfold 1,3,4, Sophie Bergmann1,3,4 & Thorsten E. Boroviak 1,3,4✉

Mammalian embryogenesis relies on glycolysis and oxidative phosphorylation to balance the generation of biomass with energy production. However, the dynamics of metabolic regulation in the postimplantation embryo in vivo have remained elusive due to the inaccessibility of the implanted conceptus for biochemical studies. To address this issue, we compiled single-cell embryo profiling data in six mammalian species and determined their metabolic dynamics through glycolysis and oxidative phosphorylation associated gene expression. Strikingly, we identify a conserved switch from bivalent respiration in the late blastocyst towards a glycolytic metabolism in early gastrulation stages across species, which is independent of embryo implantation. Extraembryonic lineages followed the dynamics of the embryonic lineage, except visceral endoderm. Finally, we demonstrate that in vitro primate embryo culture substantially impacts metabolic gene regulation by comparison to in vivo samples. Our work reveals a conserved metabolic programme despite different implantation modes and highlights the need to optimise postimplantation embryo culture protocols.

[1] Department of Physiology, Development and Neuroscience, University of Cambridge, Downing Site, Cambridge CB2 3EG, UK. [2] Wellcome Trust/Cancer Research UK Gurdon Institute, Henry Wellcome Building of Cancer and Developmental Biology, Cambridge, UK. [3] Centre for Trophoblast Research, University of Cambridge, Downing Site, Cambridge CB2 3EG, UK. [4] Wellcome Trust – Medical Research Council Stem Cell Institute, University of Cambridge, Jeffrey Cheah Biomedical Centre, Puddicombe Way, Cambridge CB2 0AW, UK. ✉email: teb45@cam.ac.uk

The generation of complex living organisms requires coordinated large-scale construction of biomass. Similar to most cellular functions, the anabolic synthesis of biomolecules is thermodynamically unfavourable and thus coupled to adenosine triphosphate (ATP) hydrolysis as a source of free energy[1,2]. In mammalian cells, ATP production relies on the processes of glycolysis and oxidative phosphorylation (OxPhos) (Fig. 1a). Glycolysis converts the 6-carbon molecule glucose (sugar) into two 3-carbon molecules of pyruvate to generate 2 ATPs. In anaerobic conditions, pyruvate is predominantly reduced to lactate; however, in the presence of oxygen, pyruvate can be additionally further converted to water and $CO_2$ by

**Fig. 1 Metabolic module score analysis from mouse single-cell RNA-seq datasets. a** Glycolysis and OxPhos pathways. G6P glucose 6-phosphate, R5P ribose-5-phosphate, PPP phosphate pentose pathway, ETC electron transport chain. **b** Mouse embryonic development. TE trophectoderm, Hyp hypoblast, Epi epiblast, ExEct extraembryonic ectoderm, VE visceral endoderm, Gast gastrulating cells, Zy zygote. **c** Substrate consumption in mouse embryos based on[18–23]. **d** Principal component analysis (PCA) of mouse embryo samples[45,46]; proportion of variance in parentheses. **e** Boxplot graph of scores of glycolysis and OxPhos modules in cell clusters representing embryonic tissues in the mouse dataset shown in (**d**). N numbers for each group are as follows: Zy = 4, 4 cell = 14, 8 cell = 47, E3.5 = 99, E4.5 Epi = 28, E5.5 = 261, E6.5 Em = 205. The boxplots are defined by the 25th and 75th percentiles, with the centre as the median. The minima and maxima extend to the largest value until 1.5 of the interquartile range (IQR) and the smallest value at most 1.5 of IQR, respectively. **f, g** Heatmaps for scaled gene expression of the mouse dataset in (**d**) for the OxPhos and glycolysis module. **h, i** Scaled expression of *Hif1a* and *Ldha* for samples shown in (**d**).

OxPhos[3,4]. OxPhos refers to the step-wise reduction of oxygen to water via the electron transport chain in the mitochondrial membranes, generating approximately 33 ATP molecules, as defined by empirical measurements[3,5]. Interestingly, fast proliferating cells, including cancer, often favour the apparently less efficient glycolysis pathway even under aerobic conditions (Warburg effect)[1,6]. The most likely explanation for the Warburg effect is that high glycolytic flux generates a multitude of anabolic precursors to synthesise biomass, while still producing sufficient ATP for cell homoeostasis[2,4]. Metabolites from glycolysis feed into the pentose phosphate pathway (PPP), which generates ribose-5-phosphate for nucleotides and NADPH-reducing power for the nucleotide and lipid biosynthesis (Fig. 1a)[7].

Similar to other rapidly proliferating cells, pluripotent cells in the early mammalian embryo equally face the challenge to balance energy production with biosynthetic demands[8]. The murine pluripotent epiblast undergoes accelerated proliferation prior to gastrulation[9,10], shifting its metabolism from bivalent, utilising both glycolytic and OxPhos pathways, towards a predominantly glycolytic flux. This metabolic change is captured in mouse embryonic stem cells (ESCs), corresponding to the preimplantation epiblast[11–13] and epiblast derived stem cells (EpiSCs), representing a post-implantation stage of development[14–16]. Mouse ESCs are metabolically bivalent, in contrast to faster proliferating EpiSCs, which are mainly glycolytic[17]. However, metabolic measurements including oxygen consumption rate and extracellular acidification rate rely on in vitro culture and are difficult to perform in post-implantation embryos of non-rodent species, in particular human and non-human primates. Thus, the regulation of glycolysis and OxPhos in the pre- to post-implantation transition in vivo has remained poorly understood.

In contrast, preimplantation development is more accessible to metabolic profiling by photometric quantification of metabolites in the culture medium. Early studies in mice have established that pyruvate acts as the major carbon source from zygote to morula, with glucose taking over during blastocyst formation[18–23] (Fig. 1b, c). Oxygen consumption remains low throughout the earliest cleavage stages, rapidly increases during blastocyst cavitation and subsequently declines[20,23] (Fig. 1b, c). Notably, the majority of preimplantation development proceeds without growth in embryo mass. It is not before the late blastocyst stage, when the three founding populations epiblast, hypoblast and trophoblast are established, that embryonic cells gain autopoiesis[24] and the weight of the embryo increases[25].

Upon implantation, mammalian embryos substantially diverge with regard to tissue architecture and extraembryonic lineage specification[26,27]. Moreover, there are at least three different modes of mammalian embryo implantation: (i) superficial, characterised by shallow trophoblast invasion and the most common form of attachment to the uterus (e.g. pig, cow, monkey), (ii) interstitial, where the blastocyst burrows through—and is fully surrounded by—the endometrium of the uterus (e.g. human, apes, guinea pig) and (iii) eccentric, which involves localisation of the embryo into a uterine cleft followed by proliferation of the

surrounding uterine tissues (e.g. mouse, rat, hamster)[28,29]. This diversity might lead to metabolic differences due to changes in the embryo microenvironment, including oxygen tension or nutrient availability. Considering that metabolic reactions provide substrates for epigenetic modifications[30–32], the metabolic state of extraembryonic tissues may constitute pivotal for embryonic development. However, it is currently unclear how species-related differences, including mode and timing of implantation, affect the metabolism in embryonic and extraembryonic lineages.

Recent protocols for human[33–35] and non-human primate[36,37] blastocyst culture to post-implantation stages have opened new avenues to study primate embryo implantation in vitro. The challenge is now to identify the most physiological culture conditions for natural as well as stem cell-based[38–44] embryo models. However, the metabolic state of embryonic and extraembryonic lineages in vitro has remained elusive.

Here, we overcome the technical issues of metabolic profiling during mammalian embryogenesis by leveraging single-cell transcriptome data to infer metabolic regulation. We compiled a transcriptional atlas of opossum, mouse, pig, marmoset, cynomolgus monkey and human embryos and determined module scores for glycolysis and OxPhos-related gene expression in stage-matched samples. This approach faithfully recapitulated known features of metabolic regulation and enabled lineage-specific interrogation of the metabolic state across species with different implantation modes. We reveal a consistent switch from bivalent respiration to glycolysis upon embryonic disc formation prior to gastrulation and elucidate metabolic dynamics in extraembryonic tissues. Finally, we leveraged our computational analysis to interrogate in vivo versus in vitro primate embryo samples and identify profound differences in metabolic regulation.

## Results
**Module score analysis recapitulates known metabolic features in the mouse embryo.** We set out to analyse the transcriptional programmes of enzymes and regulatory genes involved in glycolysis and OxPhos by calculating module scores in Seurat (see 'Methods'). As a proof of principle, we analysed mouse single-cell transcriptome embryo datasets from zygote to gastrulation (Fig. 1d, e)[45,46]. Both zygote and 4-cell stages showed low OxPhos and glycolysis module scores, indicating low enrichment of metabolic transcripts. However, the expression of OxPhos associated genes rose from the 8-cell to the blastocyst stage and peaked around the time of implantation (Embryonic day (E) 4.5). This was followed by a module score drop upon egg cylinder formation and gastrulation (E6.5). The enrichment of OxPhos genes just before implantation tightly correlated with the previously reported high oxygen consumption in mouse blastocysts (Fig. 1b)[20]. Glycolysis module scores were initially low during most of the preimplantation period until the blastocyst stage, and then steadily rose until gastrulation. These expression kinetics faithfully recapitulate the well-established increase in glucose consumption during blastocyst formation in mouse (Fig. 1b)[18–23].

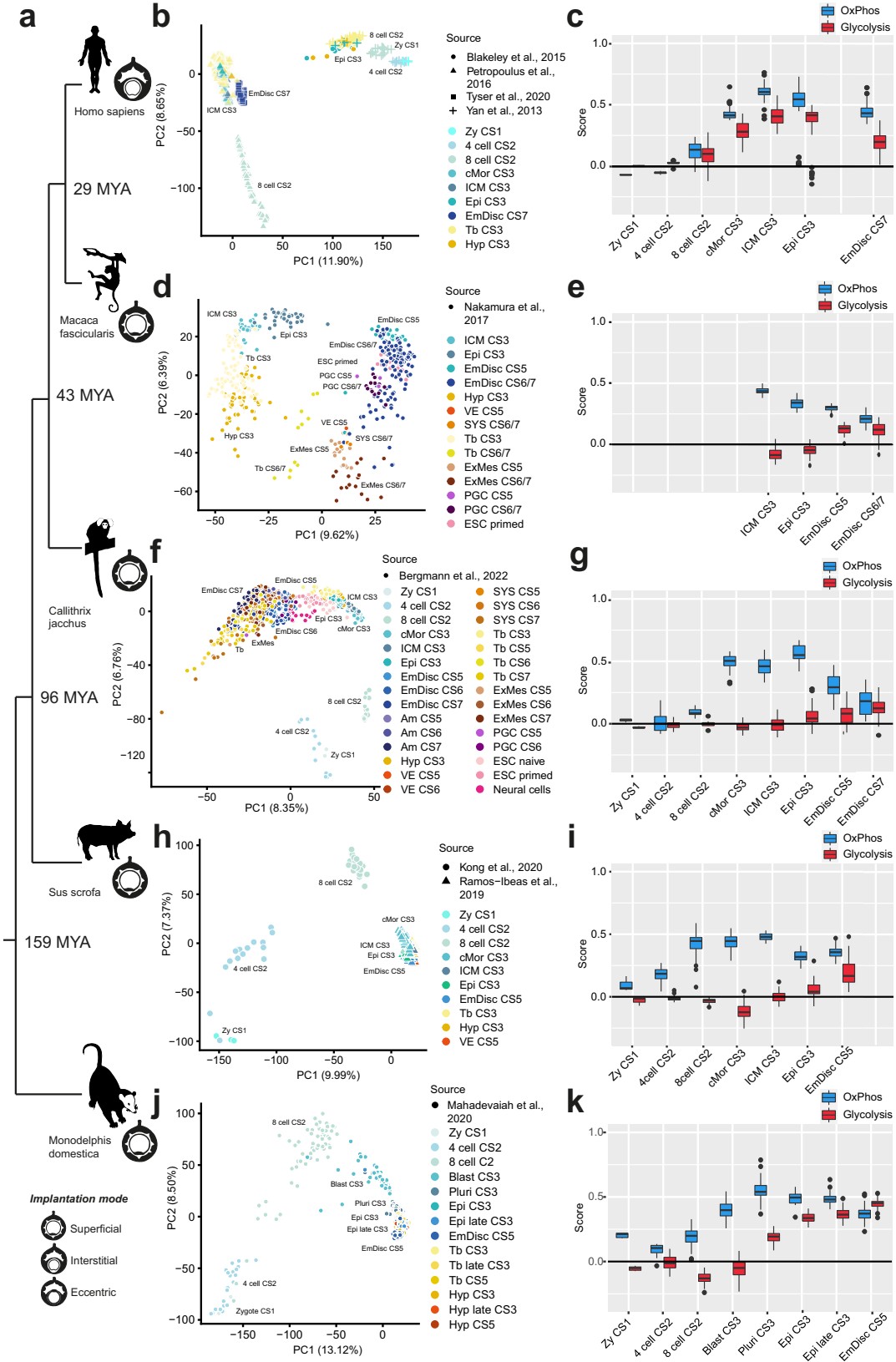

We verified that absolute gene expression levels consistently reflected the scores for OxPhos and glycolysis modules (Fig. 1f, g) and visualised metabolic gene interactions (Supplementary Fig. 1a, b). Specifically, we observed profound expression of OxPhos genes from all four complexes of the electron transfer chain and ATP synthase in the blastocyst and post-implantation stages. Glycolysis enzymes and related regulatory genes showed increased expression only during and after embryo implantation. Importantly, hypoxia-inducible factor 1 alpha (*Hif1a*), which is sufficient to switch energy metabolism from bivalent to exclusively glycolysis[17], peaked in the early post-implantation epiblast at E5.5 (Fig. 1h). Lactate dehydrogenase (*Ldha* was equally upregulated in the post-implantation stages (Fig. 1i and

**Fig. 2 The transcriptional metabolic programme in human, cynomolgus, marmoset, porcine and opossum embryonic development. a** A phylogenetic tree of five mammalian species analysed in this study. Distances are not to scale, timeline based on the Timetree tool[85]. Icons represent implantation modes in given species. MYA million years ago; values indicate time separating human from a given species. **b** PCA of samples from the human compiled dataset[50–52]. *N* numbers of individual groups are as follows: Zy CS1 = 3, 4 cell CS2 = 10, 8-cell CS2 = 94, cMor CS3 = 29, ICM CS3 = 66, Epi CS3 = 54, Epi CS7 = 147. **d** PCA of cynomolgus monkey samples[55]. *N* numbers of individual groups are as follows: ICM CS3 = 29, Epi CS3 = 68, EmDisc CS5 = 26, EmDisc CS6/7 = 129. **f** PCA of marmoset samples (Bergmann et al.[56]). *N* numbers of individual groups are as follows: Zy CS1 = 3, 4-cell CS2 = 12, 8-cell CS2 = 15, cMor CS3 = 53, ICM CS3 = 42, Epi CS3 = 38, EmDisc CS5 = 24, EmDisc CS6 = 46, EmDisc CS7 = 33. **h** PCA of porcine samples[57,58]. *N* numbers of individual groups are as follows: Zy CS1 = 3, 4-cell CS2 = 14, 8-cell CS2 = 26, cMor CS3 = 47, ICM CS3 = 24, Epi CS3 = 25, EmDisc CS5 = 48. (j) PCA of opossum samples[59]. *N* numbers of individual groups are as follows: Zy CS1 = 7, 4-cell CS2 = 21, 8-cell CS2 = 62, Blast CS3 = 55, Pluri CS3 = 140, Epi CS3 = 26, Epi late CS3 = 51, EmDisc CS5 = 106. **c**, **e**, **g**, **i**, **k** Boxplot graphs of scores for glycolysis and OxPhos modules in cell clusters representing embryonic tissues in datasets shown to the left of each graph. The boxplots are defined by the 25th and 75th percentiles, with the centre as the median. The minima and maxima extend to the largest value until 1.5 of the interquartile range (IQR) and the smallest value at most 1.5 of IQR, respectively. CS Carnegie stage, ZY zygote, 4-cell 4-cell stage, 8-cell 8-cell stage, cMor compacted morula, ICM inner cell mass, Epi epiblast, EmDisc embryonic disc, Tb trophoblast, Hyp hypoblast, VE visceral endoderm, AM amnioid, SYS secondary yolk sac, PGC primordial germ cells, ExM extraembryonic mesoderm, Blast blastocyst, Pluri pluriblast.

Supplementary Fig. 1a), consistent with a predominantly glycolytic metabolism where pyruvate is converted to lactate instead of being transported into the mitochondria for oxidation (Fig. 1a and Supplementary Fig. 1a).

To independently test Seurat-derived glycolysis and OxPhos module scores, we calculated expression scores with the VISION tool[47] and the REACTOME database ('Glycolysis' and 'Respiratory electron transport')[48]. The metabolic trends observed from VISION and REACTOME were almost identical to Seurat-derived module scores (Supplementary Fig. 1c). We conclude that glycolysis and OxPhos module score analysis is capable of inferring changes in energy metabolism from single-cell RNA-seq data.

**Metabolic dynamics in the pre- to post-implantation transition.** To gain insight into the transcriptional programmes of metabolic genes, we analysed embryo datasets of opossum, pig, marmoset, cynomolgus and human (Supplementary Data 1 and 'Methods'). We realigned 7411 full-length single-cell transcriptomes from 15 datasets to generate a stage-matched compendium of mammalian development. The combined dataset included at least one mammalian species representative for each principal mode of implantation[29,49] (Fig. 2a). The six datasets span from zygote to gastrula (Carnegie Stage (CS) 1–7, Fig. 2b, d, f, h, j) with varying degrees of evolutionary distance to humans (Fig. 2a), which allowed us to calculate metabolic module scores for 119 in vivo lineages or in vitro samples (Supplementary Data 1 and Supplementary Figs. 2–4).

The human datasets covered zygote (CS1) to late blastocyst (CS3) (Fig. 2b), with an additional timepoint at the late gastrula stage (CS7)[50–53]. OxPhos scores largely followed the dynamics in mice (Fig. 1e), with the highest expression scores being observed during blastocyst formation, followed by a decrease towards the gastrula stage (Fig. 2c). Notably, glycolysis-related expression increased slightly earlier in humans than in mice (Fig. 2c). The earliest samples for the cynomolgus monkey (Old World monkey) constituted the early blastocyst (CS3) and the datasets ended at the late gastrula stage (CS7)[54,55]. Metabolic scores were similar to trends in mouse and human, with the blastocyst stages exhibiting highest OxPhos expression and glycolysis expression increasing post implantation (Fig. 2d, e). The most complete dataset was from the marmoset monkey (New World monkey)[56] with transcriptomes spanning from zygote (CS1) to gastrula (CS7). OxPhos expression scores peaked in the preimplantation epiblast (CS3), whereas glycolysis increased from the late blastocyst stage and continuously rose until gastrulation, in accordance with the other primate species.

To delineate how embryo implantation dynamics affect embryonic metabolism, we performed metabolic inference on pig[57,58] (Fig. 2h, i) and opossum[59] single-cell embryo profiling datasets (Fig. 2j, k). In both species, embryonic disc formation and gastrulation commences prior to embryo implantation[26,57–59]. Pig embryos form spheroidal blastocysts with a flat oval embryonic disc at E10-11[58], (equivalent to CS5) and implant around E12–14[60]. The opossum belongs to the clade of marsupials and is characterised by late embryo implantation (E10–12.5) and a very short gestation period of merely 15 days[59,61,62]. Opossum and pig datasets (Fig. 2h, j) spanned from zygote to early embryonic disc stages (CS5 equivalent) and we calculated OxPhos and glycolysis module scores (Fig. 2i, k). Strikingly, OxPhos peaked at the blastocyst stage (CS3) and glycolysis gradually increased from the blastocyst until the embryonic disc stage (CS5), similar to human, monkey and mouse (Fig. 2i, k). We also examined pentose phosphate pathway associated transcripts in all samples and found a modest increase at the later gastrula stages for the majority of species analysed (Supplementary Fig. 5a).

Collectively, metabolic inference in six species suggests that the shift from bivalent to glycolytic metabolism observed in early rodent development is driven by an evolutionary conserved programme, rather than embryo implantation.

**Visceral endoderm exhibits lower metabolic scores.** Physical measurements of metabolites in isolated extraembryonic tissues of the conceptus are technically challenging, which has prevented the elucidation of lineage-specific characteristics of metabolic activity. Metabolic inference from single-cell embryo profiling datasets provides the advantage that embryonic and extraembryonic lineages can be analysed separately. We considered whether embryonic and extraembryonic tissues differ in their expression of metabolic enzymes and related genes and focused our comparisons on the late blastocyst stage as well as on visceral endoderm and the embryonic disc for post-implantation stages.

In mouse, the OxPhos scores of hypoblast and visceral endoderm samples at blastocyst and gastrulation stages declined in the pre- to post-implantation embryo transition (Fig. 3a), similarly to the embryonic lineage. Direct comparison of OxPhos scores in preimplantation epiblast and hypoblast at E4.5 showed a small reduction in the hypoblast (Fig. 3a). Glycolytic scores did not differ in the hypoblast compared to the epiblast (Supplementary Fig. 5b). However, the hypoblast-derivative visceral endoderm exhibited a marked reduction in glycolysis scores in the post-implantation embryo at E.6.5 (Fig. 3a). Human preimplantation blastocysts showed slightly reduced OxPhos-related gene expression in the hypoblast, while changes in glycolysis scores were insignificant (Fig. 3b and Supplementary

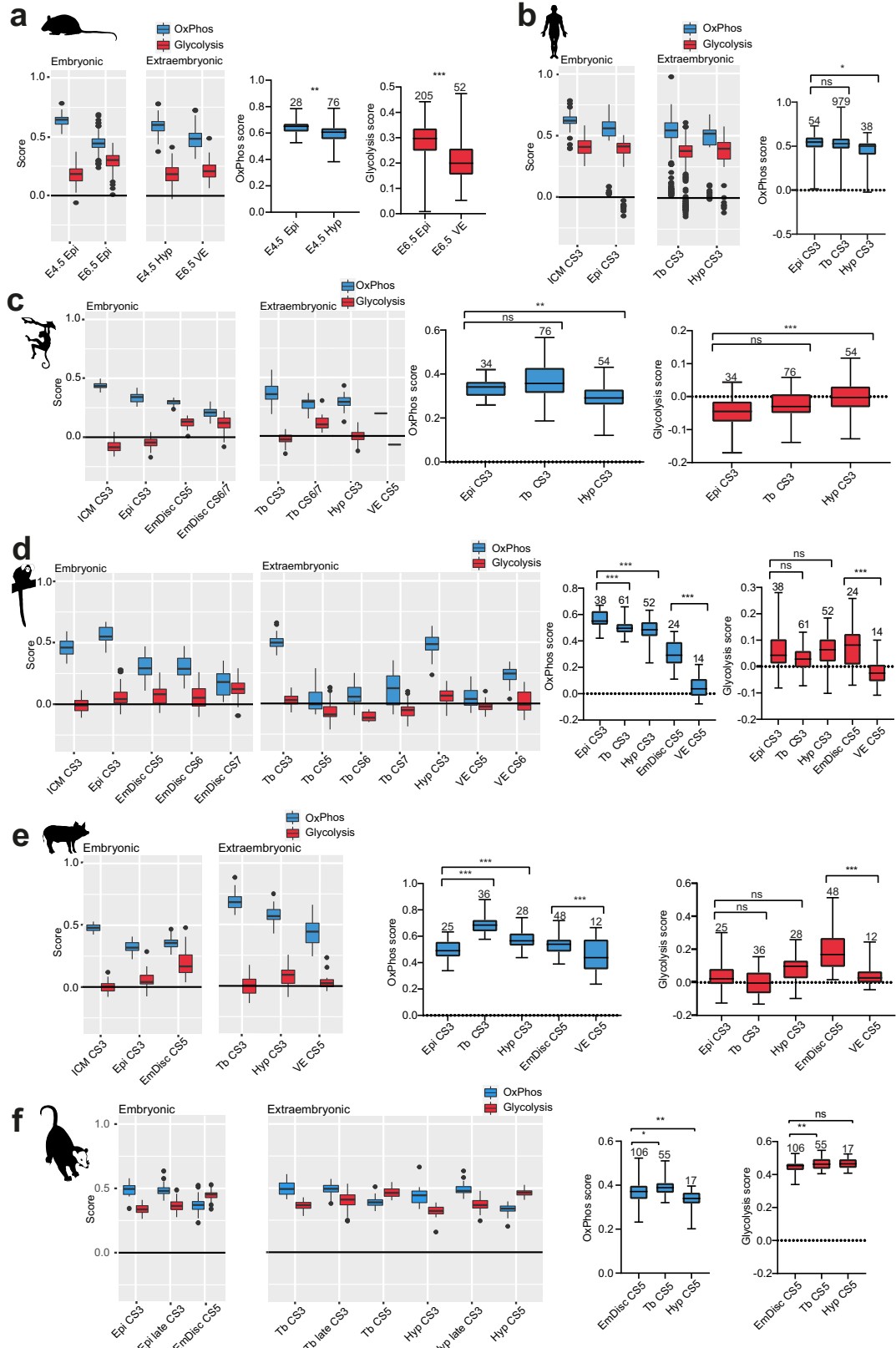

Fig. 5c). In the cynomolgus dataset, trophoblast did not differ from epiblast in either OxPhos or glycolysis scores (Fig. 3c). Marmoset late blastocysts glycolysis scores were not significantly different between embryonic and extraembryonic tissues, however, OxPhos module scores were reduced in trophoblast and hypoblast, compared to epiblast (Fig. 3d). In the post-implantation embryo, visceral endoderm exhibited substantially lower expression of OxPhos and glycolysis modules

**Fig. 3 Comparison of metabolic gene enrichment in embryonic and extraembryonic tissues. a** Boxplot graphs for glycolysis and OxPhos module scores of mouse embryonic and extraembryonic tissues, comparisons tested using two-sided Student's $t$ test. $P$ values are as follows: Epi 4.5 vs Hyp 4.5 = 0.003, Epi E6.5 vs VE E6.5 = 0.001. **b** Module scores for human samples, comparisons tested using Kruskall–Wallis test. $N$ numbers for groups (apart from ones shown) are as follows: ICM CS3 = 66. $P$ values are as follows: Epi CS3 vs Tb CS3 = 0.114, Epi CS3 vs Hyp CS3 = 0.002. **c** Module scores for cynomolgus monkey samples, comparisons tested using Kruskall–Wallis test for OxPhos and one-way ANOVA for glycolysis. $N$ numbers for groups (apart from ones shown) are as follows: ICM CS3 = 66, Tb CS6/7 = 11, VE CS5 = 1. $P$ values for OxPhos score are as follows: Epi CS3 vs Tb CS3 = 0.114, Epi CS3 vs Hyp CS3 = 0.002, for glycolysis score: Epi CS3 vs Tb CS3 = 0.067, Epi CS3 vs Hyp CS3 < 0.001. **d** Module scores for marmoset samples, comparisons tested with one-way ANOVA for OxPhos and Kruskall–Wallis test for glycolysis. $N$ numbers for groups (apart from ones shown) are as follows: ICM CS3 = 42, EmDisc CS6 = 46, EmDisc CS7 = 33, Tb CS5 = 39, Tb CS6 = 44, VE CS6 = 16. $P$ values for OxPhos score are all <0.001, for glycolysis score: Epi CS3 vs Tb CS3 = 0.143, Epi CS3 vs Hyp CS3 = 0.999, EmDisc CS5 vs VE CS5 = 0.003. **e** Module scores for porcine samples, tested with Kruskall–Wallis test for OxPhos and one-way ANOVA for glycolysis. $N$ numbers for groups (apart from ones shown) are as follows: ICM CS3 = 24. $P$ values for OxPhos scores are all <0.001, for glycolysis scores: Epi CS3 vs Tb CS3 = 0.81, Epi CS3 vs Hyp CS3 = 0.163, EmDisc CS5 vs VE CS5 = 0.001. **f** Module scores for opossum samples, tested with one-way ANOVA. $N$ numbers for groups (apart from ones shown) are as follows: Epi CS3 = 26, Epi late CS3 = 51, Tb CS3 = 46, Tb late CS3 = 64, Hyp CS3 = 14, Hyp late CS3 = 33. $P$ values for OxPhos scores are: EmDisc CS5 vs Tb CS5 = 0.001, EmDisc CS5 vs Hyp CS5 = 0.075, for glycolysis scores: EmDisc CS5 vs Tb CS5 = 0.010, EmDisc CS5 vs Hyp CS5 = 0.012. ns nonsignificant, \*$P < 0.05$, \*\*$P < 0.01$, \*\*\*$P < 0.001$. The boxplots on the left are defined by the 25th and 75th percentiles, with the centre as the median. The minima and maxima extend to the largest value until 1.5 of the interquartile range (IQR) and the smallest value at most 1.5 of IQR, respectively. In the boxplots on the right the maxima and minima refer to the largest and smallest values, respectively. In all cases, Kruskall–Wallis tests were followed by Dunn's multiple comparisons test and ANOVA's were followed by Sidak's multiple comparisons test. CS Carnegie stage, ZY zygote, 4-cell 4-cell stage, 8-cell 8-cell stage, cMor compacted morula, ICM inner cell mass, Epi epiblast, EmDisc Embryonic disc, Tb trophoblast, Hyp hypoblast, VE visceral endoderm.

in comparison to the embryonic disc. We made the same observation in pig, where both OxPhos and glycolysis associated transcripts were significantly lower in the visceral endoderm in comparison to the early embryonic disc at CS5 (Fig. 3e). The opossum followed the same trend for OxPhos in the hypoblast at E7.5 (equivalent to CS5 VE), but not glycolysis (Fig. 3f and Supplementary Fig. 5d). Taken together, the transcriptional dynamics of OxPhos and glycolysis in vivo reveal that extraembryonic tissues largely follow the embryonic lineage. Visceral endoderm at CS5 constitutes an exception, which showed an overall reduction in OxPhos and glycolysis in mouse, marmoset, pig and for OxPhos in the opossum.

**In vitro culture affects metabolic gene expression.** To examine the effect of in vitro culture on OxPhos and glycolysis expression scores, we incorporated single-cell transcriptomes from embryo-derived stem cell lines into the six species embryo dataset. We compared mouse embryonic stem cells (ESCs) cultured in the presence of MEK and GSK-3b inhibition plus leukaemia inhibitory factor (2i + LIF) corresponding to the naive preimplantation epiblast, ESCs maintained in serum plus LIF (Serum + LIF) and Epiblast Stem Cells (EpiSCs) cultivated in ActA/bFGF as representatives of the primed post-implantation epiblast[11,12,14]. Embryo-derived cells separated according to culture condition and developmental stage (Fig. 4a)[63]. Transcriptional metabolic scores indicated that OxPhos is highest in ESCs in 2i+LIF, while glycolysis peaked in EpiSCs (Fig. 4b, c). ESCs in Serum+LIF are more heterogenous and partially exit naive pluripotency[12,64,65]. Consistent with his notion, OxPhos and glycolysis scores were in between ESCs in 2i+LIF and EpiSCs (Fig. 4b, c).

We next set out to assess metabolic scores in naive and primed human pluripotent stem cells (hPSCs)[66]. Naive hPSCs exhibited lower glycolysis scores and higher OxPhos scores than primed hPSCs (Fig. 4d, f). Metabolic scores of primed hPSCs did not change after ten passages in culture (Supplementary Fig. 5e). Overall, glycolysis scores were substantially higher in human compared to mouse, which may indicate a greater disposition toward the glycolytic pathway or effects from the culture medium.

A recent report from Xiang et al. described the consistent generation of human post-implantation stages by in vitro 3D culture using Matrigel[35]. We extended our comparisons to

include the Xiang et al. single-cell RNA-seq dataset of human post-implantation culture spanning from late blastocyst (CS3) to early gastrula (CS6)[35] (Fig. 4g). Interestingly, we observed elevated glycolytic scores throughout developmental stages and lineages (Fig. 4h), which led us to ask whether primate embryos change their metabolism upon in vitro culture. To test this hypothesis, we included a cynomolgus monkey in vitro dataset using similar culture conditions[36] (Fig. 4i, j) and evaluated metabolic expression scores against in vivo cynomolgus samples[54] (Fig. 4k). We found that both OxPhos and glycolysis scores are higher in embryonic discs of embryos cultured in vitro at CS5 and CS6 (Fig. 4j, k). This result demonstrates that current in vitro culture protocols of primate embryos generally increase metabolic scores, independent of lineage or developmental stage.

## Discussion

In this study, we leveraged module score analysis to infer metabolic dynamics during embryogenesis in six mammalian species. We show that all six mammalian embryos undergo a conserved transcriptional transition of metabolic gene expression from bivalent towards predominantly glycolytic metabolism at the time of embryonic disc formation (Fig. 5). This metabolic switch occurs despite an evolutionary distance of 159 million years as well as different mechanisms and time points of embryo implantation. Furthermore, different species initiate gastrulation at a different time, with primitive streak formation occurring as early as E6.5 in the mouse and as late as E14–16 in the marmoset[45,56]. Mammalian embryogenesis is therefore characterised by an intrinsic metabolic programme, conserved regardless of embryo size or timing of gastrulation[54,58,62,67,68].

Increase in OxPhos after cleavage is indicative of booting of the metabolic machinery in the embryo to meet the energy demands for cell re-arrangements and lumen formation during blastocyst formation[69,70]. Glycolytic activity is initiated thereafter, when embryonic cells become autopoietic in the preimplantation epiblast[12,24] and continuously increases during early post-implantation development. The shift toward glycolysis likely constitutes a critical boost to the biosynthetic capacity of embryonic cells as they enter the most rapid growth phase in the life of the embryo[1,71], similar to the Warburg effect in proliferating cancer cells[1,6]. It has been suggested that the Warburg effect channels metabolites into the PPP to synthesise essential

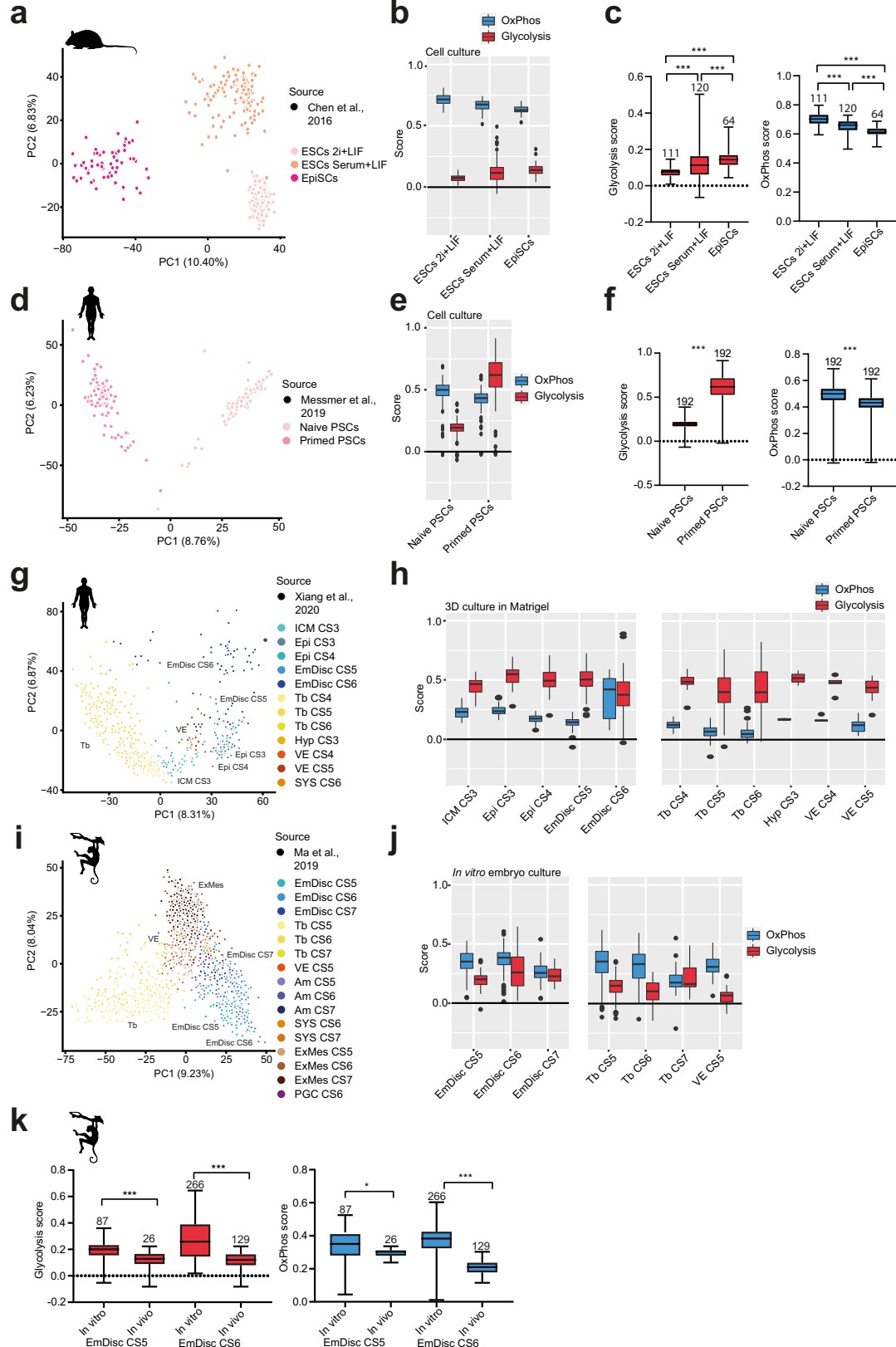

macromolecules and create new biomass[8]. However, we observed only moderate changes in the expression of PPP-associated genes. It is conceivable that PPP enzymes might be regulated at the protein level, either post-transcriptionally or through substrate availability. Quantitative physiological measurements will be required to delineate the regulatory mechanisms of the PPP in embryonic development.

Computational inference of OxPhos and glycolysis dynamics offered an efficient approach to assess embryonic and extra-embryonic lineages in vivo. Our analysis demonstrates that

**Fig. 4 OxPhos and glycolysis module score expression in human embryonic cell and primate embryo culture in vitro. a** PCA of mouse embryonic stem cells[63]. **b** Glycolysis and OxPhos module scores of samples shown in (**a**). **c** Comparisons of glycolysis and modules scores tested with Kruskal–Wallis test. P values for glycolysis scores are as follows: ESCs 2i+LIF vs ESCs Serum <0.001, ESCs 2i+LIF vs EpiSCs <0.001, ESCs Serum vs EpiSCs=0.001., for OxPhos scores all P values are <0.001. **d** PCA of human pluripotent stem cells[66]. **e** Module scores for samples in (**d**). **f** Comparisons of module scores tested using Mann–Whitney test. P values are as follows: for glycolysis <0.001, for OxPhos <0.001. **g** PCA of human embryo samples in 3D culture[35]. **h** Module scores of samples in (**g**). N numbers are as follows: ICM CS3 = 28, Epi CS3 = 27, Epi CS4 = 33, EmDisc CS5 = 59, EmDisc CS6 = 69, Tb CS4 = 27, Tb CS5 = 215, Tb CS6 = 65, Hyp CS3 = 2, VE CS4 = 6, VE CS5 = 17. **i** PCA of cynomolgus monkey post-implantation in vitro cultured embryo samples[36]. **j** Modules scores of samples in (**i**). N numbers, apart from the ones shown in (**k**) are as follows: EmDisc CS7 = 34, Tb CS5 = 294, Tb CS6 = 65, Tb CS7 = 21, VE CS5 = 45. **k** Comparisons of module scores between dataset in (**i**) and samples of stage-matched embryos developed in vivo[55], tested by Kruskal–Wallis test. P values for glycolysis are <0.001, for OxPhos: In vitro EmDisc CS5 vs In vivo EmDisc CS5 = 0.491, In vitro EmDisc CS6 vs In vivo EmDisc CS6/7 < 0.001. ns nonsignificant, *P < 0.05, **P < 0.01, ***P < 0.001. The boxplots on the left are defined by the 25th and 75th percentiles, with the centre as the median. The minima and maxima extend to the largest value until 1.5 of the interquartile range (IQR) and the smallest value at most 1.5 of IQR, respectively. In the boxplots on the right the maxima and minima refer to the largest and smallest values, respectively. In all cases, Kruskal–Wallis tests were followed by Dunn's multiple comparisons test and ANOVA's were followed by Sidak's multiple comparisons test. CS Carnegie stage, ZY zygote, 4-cell 4-cell stage, 8-cell 8-cell stage, cMor compacted morula, ICM inner cell mass, Epi epiblast, EmDisc embryonic disc, EpiSCs epiblast-like stem cells, PSCs pluripotent stem cells, ESCs embryonic stem cells, Tb trophoblast, Hyp hypoblast, VE visceral endoderm, AM amnioid, SYS secondary yolk sac, PGC primordial germ cells, ExM extraembryonic mesoderm.

extraembryonic tissues largely follow the embryonic lineage, except for visceral endoderm at CS5. The reduction of OxPhos and glycolysis module scores in visceral endoderm may be a consequence of advanced differentiation or slower proliferation compared to the embryonic disc. Unfortunately, human and cynomolgus datasets did not contain samples reliably annotated as visceral endoderm and further studies will be needed to delineate specific roles of the metabolism in extraembryonic tissues.

Finally, we provided insights into the energy metabolism of in vitro models of rodent and primate embryogenesis. Inference of metabolic dynamics robustly captured the well-established differences between mouse naive pluripotent ESCs and primed EpiSCs, which exhibit lower mitochondrial activity and a higher glycolytic rate compared to ESCs[17]. Moreover, we interrogated the latest protocols for cynomolgus post-implantation embryo culture[35,36] and identified profoundly increased metabolic scores compared to embryonic lineages in vivo. One possible reason for this result could have been differences in oxygen accessibility. The uterine environment is hypoxic[72] and the embryos analysed here were cultured under atmospheric oxygen conditions[36] (see Supplementary Data 1). This would predict lower glycolysis scores, as hypoxia of the embryo microenvironment would induce glycolytic metabolism[17,73]. However, we observed increased glycolytic scores for cynomolgus embryos cultured in vitro. A more likely explanation maybe that metabolic changes are caused by the composition of embryo culture media, as substrate availability may influence metabolic flux[71,74]. Current culture conditions may either lack certain substrates or contain excessive levels of key metabolites, including glucose, and thus induce changes in the metabolic programme.

Collectively, this work identifies a conserved metabolic programme in early mammalian embryogenesis and highlights differences in metabolic gene regulation between in vivo versus in vitro developed primate embryos. Our metabolic compendium provides a rich resource for comparative embryology and an avenue for the development of more physiological media for embryo culture and assisted reproductive technologies.

## Methods

**Dataset assembly**. A total of fifteen separate single-cell RNA-seq datasets were compiled and stage-matched to create a collection of transcriptomic data in pre- and early post-implantation stages for six mammalian species. The datasets were analysed in R studio using R software v4.00 (www.r-project.org). Seurat v3 platform was used for single-cell RNA-seq data analysis[75]. A list of all used datasets with accession codes is presented in Supplementary Data 1. Principal component

analysis (PCA) using Seurat was performed using the 20,000 most variable genes. Merging was performed using Seurat's 'merge' function. Merged or single datasets as indicated by PCA plots in Figs. 1, 2 and 4 were log-normalised before performing any analysis. Cells from the Deng et al.[46] (*Mus musculus*, zygote to E2.5) dataset were filtered based on a previously published sc-RNA-seq meta-analysis[76]. The Mohammed et al. dataset (*Mus musculus*, from E3.5) was re-clustered and the cell types were re-annotated with Seurat due to lack of available metadata[45]. Cell clusters of the two pig datasets[57,58] (*Sus scrofa*) were renamed based on key developmental and morphological features to match the Carnegie developmental staging used in primates. The human dataset (*Homo sapiens*) in Fig. 2 was compiled using cells from three datasets[50–52] that were filtered and re-annotated based on a previous publication[77], merged with re-analysed and filtered cells from the human gastrula dataset[53]. The rest of datasets were analysed separately or merged as indicated by PCA plots. The two pig datasets were analysed separately due to a large discrepancy in the number of available features (genes) between the two datasets.

**Gene modules**. OxPhos and glycolysis-related gene lists were compiled based on Gene Ontology entries for 'oxidative phosphorylation' and 'canonical glycolysis', filtered for mammalian genes[78,79]. Afterwards, additional genes were added to both lists based on glycolysis and OxPhos markers used in the literature for metabolic studies[4,17,80]. Gene sets were filtered against all features in the single-cell RNA-seq expression matrix for each dataset, generating species-specific gene lists used for further analysis. On average, gene sets contained 130 and 30 genes for OxPhos and glycolysis, respectively. Interaction map for glycolysis and OxPhos gene sets in mouse (Supplementary Fig. 1a, b) were created using Cytoscape[81] based on WikiPathways diagrams for 'Glycolysis' and 'Electron Transport Chain' retrieved using the Pathway Commons plugin[82,83].

**Module score and statistical analysis**. The 'AddModuleScore' function from the Seurat v3 platform was used to calculate the expression score for the glycolysis and OxPhos gene sets. Module score function allows calculation of average expression of genes in a specific module on a single-cell level, from which a control cell score is subtracted. The control cell scores are calculated based on binning all genes based on average expression and randomly selecting a given number of genes from each bin. Therefore, a positive module score means that the expression of a given set of genes is increased in a cell in comparison to a randomly assembled gene set. Default settings were applied with number of bins equal to 24 and control size from a bin equal to 100. For more in-depth description of this analysis, see ref. [84].

Statistical tests of differences in module score as mentioned in the text or figure legends were performed in GraphPad Prism version 8.4.2, with appropriate tests applied based on normality testing (Shapiro–Wilk and Kolmogorov-Smirnov tests). All *t* tests are two-sided. Sidak's multiple comparisons test and Dunn's multiple comparisons test were used for posthoc analysis of the ANOVA's and the Kruskal–Wallis tests, respectively. For comparison with the module score results, we produced gene expression heatmaps for all tested datasets in Supplementary Figs. 2–4.

**Single-gene expression analysis**. The expression of *Hif1a* and *Ldha* in mouse dataset (Fig. 1h) was calculated using Seurat v3 log-normalisation and scaling.

**Calculating scores using VISION**. The VISION pipeline was run using RNA count data and the signature (gene set) scores were obtained using the 'analyse'

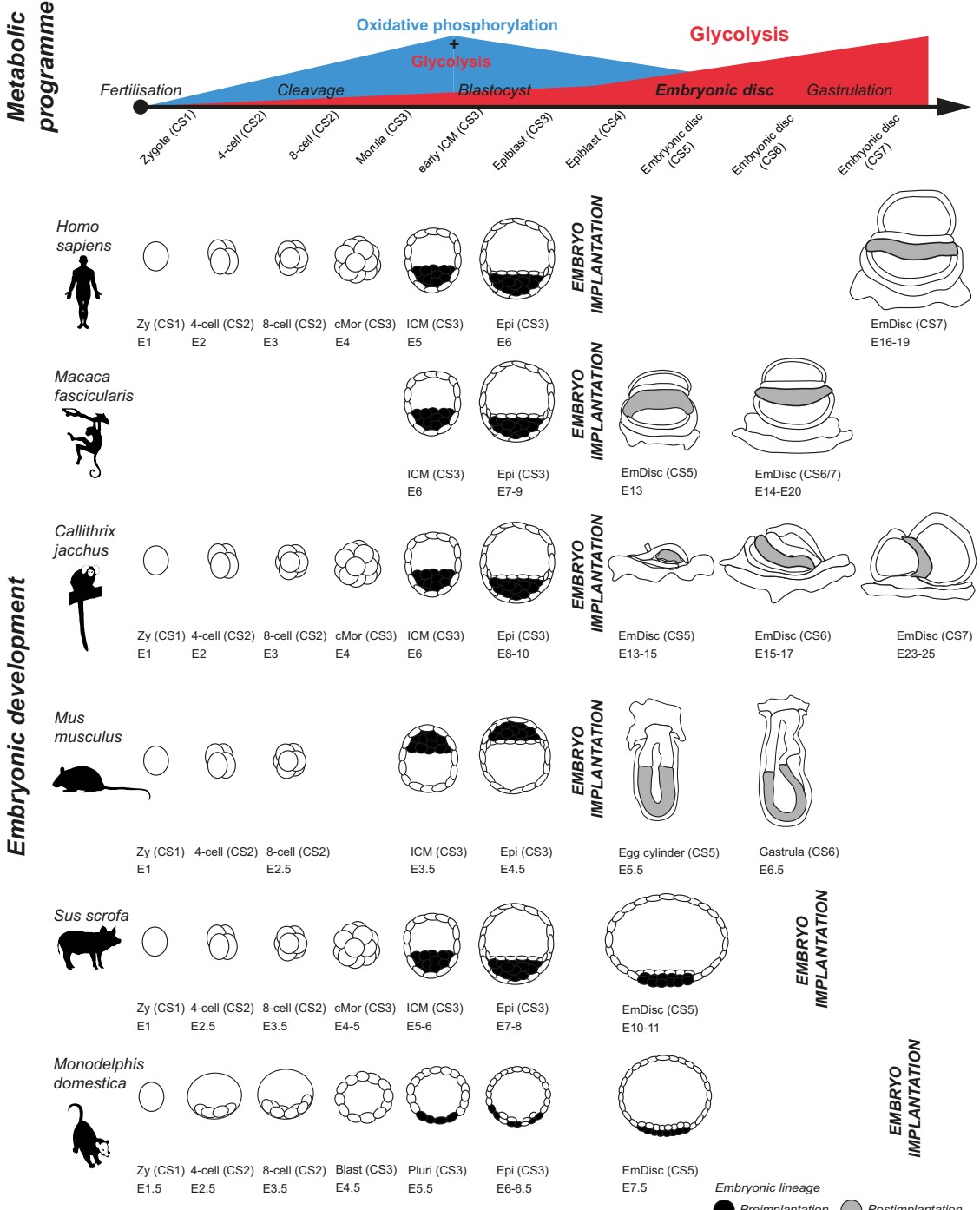

**Fig. 5 An evolutionary conserved switch from bivalent to glycolytic metabolism in mammalian development upon embryonic disc formation.** Single-cell RNA-seq data from six species stage-matched according to the Carnegie system allowed elucidation of the transcriptional programme of metabolic enzymes. This metabolic programme is characterised by low glycolytic activity during cleavage, followed by a peak in OxPhos during the formation of the blastocyst (CS3) and an increase in glycolytic flux at the time of embryonic disc formation (CS5), regardless of species or time of implantation. CS Carnegie stage, Zy zygote, cMor compacted morula, ICM inner cell mass, Epi epiblast, EmDisc embryonic disc, Blast blastocyst, Pluri pluriblast, E embryonic day.

function[47]. Unsigned signatures were either defined manually or derived from the REACTOME database[48] using MSigDB depository as.gmt files.

**Data visualisation**. Boxplots with medians of the module scores in cell clusters were created using the ggplot2 package (Figs. 1, 2, 3 and 4b, e, h, j) or GraphPad Prism v9.1.0 (Figs. 3 and 4c, f, k). The hinges of the boxplot are defined by the first and third quartiles (the 25th and 75th percentiles). In ggplot boxplots, the upper whisker extends to the largest value until 1.5 of the interquartile range (IQR) and the lowest whisker extends to the smallest value at most 1.5 of IQR. Outlying points are plotted. In the GraphPad plots, the whiskers extend to the smallest and largest values. PCA plots and gene expression heatmaps for comparison with the module score were plotted with Seurat using log-normalised and scaled data. Figures were assembled using Adobe Illustrator 25.2.2 and all images (including species icons, pathway diagrams and embryo drawings) were created by the authors using Illustrator.

**Reporting summary**. Further information on research design is available in the Nature Research Reporting Summary linked to this article.

## Data availability

The datasets used in this study are publicly available under accession codes supplied in Supplementary Data 1. All other relevant data supporting the key findings of this study are available within the article and its Supplementary Information files or from the corresponding author upon reasonable request. Source data are provided with this paper.

## Code availability

The code used to calculate module score based on mammalian glycolysis and OxPhos gene lists is available on GitHub under the following link: https://github.com/Boroviak-Lab/HexaSpeciesMetabolomicsAnalysis.git, https://doi.org/10.5281/zenodo.5733306.

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

## Acknowledgements

We would like to thank the members Boroviak lab for their enthusiasm and critical discussion of the manuscript, in particular Erin Slatery. This research is generously supported by the Wellcome Trust (WT RG89228) and the Centre for Trophoblast Research. A.M. is funded by the Wellcome Trust Developmental Mechanisms PhD programme. T.E.B. is a Wellcome Trust—Royal Society Sir Henry Dale Fellow.

## Author contributions

A.M. and C.P. carried out the computational analysis, prepared figures and helped with manuscript preparation. S.B. generated post-implantation marmoset embryo profiling data and provided comments on the manuscript. A.M. and T.E.B. wrote the manuscript; A.M., C.P. and T.E.B. conceived the project.

## Competing interests

The authors declare no competing interests.

## Additional information

**Peer review information** *Nature Communications* thanks Roger Sturmey and the other anonymous reviewer(s) for their contribution to the peer review this work. Peer reviewer reports are available.

