## [Peer Review File · Nature Communications]

Reviewers' Comments:

Reviewer #1:

Remarks to the Author:

This is a well written account of a piece of work to compare the expression patterns of genes related to energy metabolism through the pre- and early post implantation stages in mammalian embryos. Using largely published data sets obtained from 5 different species, the authors have carried out an analysis of the way in which genes related to glycolysis and oxidative phosphorylation vary and have found that the expression profile of genes involved in these pathways largely follows the known profiles of substrate depletion and metabolic function.

As mentioned above, the manuscript is very well written and easy to read. The introduction begins with a brief account of energy metabolism, however, as a consequence of brevity, possible inaccuracies have resulted. For example, the authors state:

"in anaerobic conditions pyruvate is reduced to lactate; however in the presence of oxygen...", however this statement is an oversimplification, since a core feature of embryo metabolism (and indeed many other cells) is the concept of aerobic glycolysis, whereby even in the presence of oxygen, lactate can (and is) still produced. In addition the authors later state that 'up to 38 ATP molecules' can be generated from oxidative phosphorylation. This is a mathematically theoretical value but largely an overestimate. Oxphos is never 100% efficient and this doesn't account for proton leakage nor energy required for ATP export. See the work of Martin Brand for a more accurate account of P:O ratios and ATP stoichiometry. Later in the intro (after (O'Farrell et al 2004). The authors refer to "This metabolic shift" but do not describe WHICH metabolic shift.

In terms of the methodology, I was surprised to see a gene ontology approach, and even more so to see that 'additional genes were added based on lists'. Using GO potentially risks missing interactions between gene networks, and the addition of 'additional genes' risks bias and also risks missing novel gene interactions and/or previously undescribed gene functions. An unbiased network approach to analysis and then refinement of a metabolic gene interactome would have been much more impactful.

The data presented are well described and figures are generated very nicely. They are clear and support the overall hypothesis, although given the wealth of data describing embryo metabolism during the preimplantation stage, this is not surprising. One question that is, in my view, not adequately addressed is regarding the expression changes that accompany "a conserved metabolic transition at the time of implantation" – this is a difficult point to make from the current set of data (notably with the inclusion of the pig) – indeed the data doesn't support the proposition that this conserved implantation switch occurs, because the pig undergoes a rapid and protracted period of elongation that does not attach until much later than the other species studied. Thus, the metabolic switch (which is interesting) is surely INDEPENDENT of implantation? Consideration of other elongating species (i.e. the bovine) would help further understand this observation. This may be relevant in the context of Figure 3, which shows that porcine TE and hypoblast retains relatively high expression of Oxphos, indicative of divergence in the regulation.

Furthermore, the authors refer to the increase in glycolysis as being a necessary boost for biosynthetic capacity; a hypothesis that is quote established (i.e. the work of Matthew Vander Heiden, who is well cited in this piece), however the authors could surely have actually answered this question within this study by looking as well as the key biosynthetic genes, such as those involved with the Pentose Phosphate Pathway (at a minimum).

The authors also describe that physical measurements of metabolites during preimplantation development requires multiple whole embryos: This statement is incorrect. A number of authors from a range of laboratories have described metabolic function of single mammalian embryos (e.g. Gardner, Sakkas, Leese, Sturmey, Krisher, Prather, Thompson, Houghton to name but a few). Indeed, the authors follow this statement by implying that metabolic inference is needed to study metabolism of embryonic and extraembryonic (i.e. TE) lineages, however this again has been done directly on at least two occasions (i.e. Gopichandran and Leese 2003 and Houghton 2006).

Finally, the authors attempt to utilise data from ESCs to describe the effects of in vitro culture on gene expression. However, in my view, what this section describes is the effect of prolonged in vitro culture on stem cells, which is of interest, but not as a proxy for embryo culture if comparing in vivo vs in vitro conditions. Indeed, a glance at the data sets used indicate that there was some considerable variation between the origins of the embryos, with some being defined as in vivo and some as in vitro derived – including all of the human embryos. This section would benefit from a more nuanced and accurate account of what is being compared, since a real direct comparison of fully invitro derived embryos with closely aligned, stage-matched embryos harvested at contemporaneous time points to understand the true effects of in vitro embryo culture would be of great value – however this is not what has been presented in this work.

Reviewer #2:

Remarks to the Author:

The central claim of the paper is "a conserved switch from bivalent respiration in the late blastocyst towards a predominantly glycolytic metabolism in early gastrulation stages across species" (abstract, p.2, l. 8ff.). The species in question are 5 mammals (mouse, pig, new and old world monkeys, and human) selected for their different (gastrulation-independent) implantation modes and the claim is based on an elaborate and well informed comparison of single-cell transcriptome profiles from equivalent embryonic stages. Additional claims concern comparisons (using datasets from single or selected species) between in vitro and in vivo growth, between embryonic compartments, and between embryonic tissues and different stem cell cultures. The author are to be commended for their timely effort to draw attention to the marked differences in early development between mammals and for combining their expertise in comparative embryology and bioinformatics. This resulted in a much desired (universal, i.e.,Carnegie) stage-related transcriptome atlas for early mammalian development and there is every reason to believe that the bioinformatics part is sound and solid. The correlation, however, of gene expression profiles with gastrulation and implantation could be rendered more convincing by a systematic description and presentation of the materials and the results and by a rigorous logic connection between published results and the study design.

Major points:

- 1) Apart from overall tissue architecture and modes of implantation, early mammalian development varies substantially with regard to size of the (vigorously growing) embryonic disc (macaque and human are less than half in size and in cell numbers compared to pig, callithrix and mouse at the beginning of gastrulation) and developmental time spent until gastrulation starts (e.g. 6.5 days in mouse, 10 days in pig, and up to 50 days in callithrix). This needs to be taken into account next to the timing (and topography) of implantation when considering proliferation and mode of ATP generation, for example.
- 2) The introduction should clarify the priority of the many problems addressed or, indeed, to be solved: improving culturing conditions, differences in in vivo/in vitro development, interspecies tissue architecture comparisons, pre- and post-implantation differences, embryonic-extraembryonic lineage differences, equivalence of various types of stem cells with embryonic lineages etc.. Full credit should be given (s. page 3, bottom half) to Krisher and Prather (2012) for their detailed discussion on the metabolic phenotypes of early mammalian embryos (mainly pig and cow, though) and the Warburg effect, while O'Farrel's (2004) needs to be addressed as dealing with mouse and rat, only, (and not with observations "throughout the mammalian kingdom").
- 3) In material and methods, species names should be given when referring to datasets (p.3, 3rd paragr.) and Table 1 should be divided into a list of (a) embryos and (b) all (stem) cell cultures that were analysed. It is not clear whether the results section on stem cell lines (p. 10) are derived from published datasets or whether datasets were created for this study. Methodology for HIF1A and LDHA should be given here (and the rationale for the analysis explained in the Introduction). Why was the cynomolgus dataset by Niu et al. (2019) published in the same issue back-to-back with Ma et al. (2019) not included (or mentioned, at least)?
- 4) Stating that "the five species ... adhered to a conserved metabolic transition at the time of

implantation" is an oversimplification at least in the case of callithrix (which implants well after the peak at CS3) and pig (which implants even later: during high-somite-number stages).

Minor points

- 1) OxPhos should be used throughout instead of using varying this term so central to the study.
- 2) Fig. 1 does not show a human dataset (p.5, 3rd paragr.).
- 3) The Results and Discussion section would be easier to understand (and probably transport the evidence more clearly) if stripped of methodological or introductory descriptions (e.g. p.7, 1st two sentences; p.7 last paragr, and p.8 1st paragr).
- 3) Bibliographical details are incomplete in high number of references.

Reviewer response

to

“A hexa-species transcriptome atlas of mammalian embryogenesis delineates metabolic regulation across implantation modes, lineages and culture conditions”

Reviewer #1 (Remarks to the Author):

This is a well written account of a piece of work to compare the expression patterns of genes related to energy metabolism through the pre- and early post implantation stages in mammalian embryos. Using largely published data sets obtained from 5 different species, the authors have carried out an analysis of the way in which genes related to glycolysis and oxidative phosphorylation vary and have found that the expression profile of genes involved in these pathways largely follows the known profiles of substrate depletion and metabolic function.

As mentioned above, the manuscript is very well written and easy to read. The introduction begins with a brief account of energy metabolism, however, as a consequence of brevity, possible inaccuracies have resulted. For example, the authors state: “in anaerobic conditions pyruvate is reduced to lactate; however in the presence of oxygen...”, however this statement is an oversimplification, since a core feature of embryo metabolism (and indeed many other cells) is the concept of aerobic glycolysis, whereby even in the presence of oxygen, lactate can (and is) still produced. In addition the authors later state that ‘up to 38 ATP molecules’ can be generated from oxidative phosphorylation. This is a mathematically theoretical value but largely an overestimate. Oxphos is never 100% efficient and this doesn’t account for proton leakage nor energy required for ATP export. See the work of Martin Brand for a more accurate account of P:O ratios and ATP stoichiometry. Later in the intro (after (O’Farrell et al 2004) the authors refer to “This metabolic shift” but do not describe WHICH metabolic shift.

We thank the reviewer for their positive assessment of our work. We agree that some statements in the introduction have been oversimplifications and have changed this accordingly. For instance, we have added “predominantly” and “can be additionally further converted” to make clear that the generation of lactate is not mutually exclusive with oxidative phosphorylation.

*“In anaerobic conditions, pyruvate is **predominantly reduced to lactate; however, in the presence of oxygen, pyruvate can be additionally further converted to water and CO₂** by OxPhos^{3,4}.”*

We included references for a more accurate account of P:O ratios in the introduction.

*“OxPhos refers to the step-wise reduction of oxygen to water via the electron transport chain in the mitochondrial membranes, **generating approximately 33 ATP molecules, as defined by empirical measurements^{3,5}**.”*

We now explain exactly which metabolic shift we refer to in the main text.

*“Similar to other rapidly proliferating cells, pluripotent cells in the early embryo equally face the challenge to balance energy production with biosynthetic demands⁷. The pluripotent epiblast undergoes accelerated proliferation prior to gastrulation⁸, **shifting its metabolism from bivalent, utilizing both glycolytic and OxPhos pathways, towards a predominantly glycolytic flux.** This metabolic change is captured in mouse embryonic stem cells (ESCs), corresponding to the preimplantation epiblast¹⁰⁻¹² and epiblast derived stem cells (EpiSCs), representing a postimplantation stage of development¹³⁻¹⁵. Mouse ESCs are metabolically bivalent, in contrast to faster proliferating EpiSCs, which are mainly glycolytic¹⁶.”*

In terms of the methodology, I was surprised to see a gene ontology approach, and even more so to see that ‘additional genes were added based on lists’. Using GO potentially risks missing interactions between gene networks, and the addition of ‘additional genes’ risks bias and also risks missing novel gene interactions and/or previously undescribed gene functions. An unbiased network approach to analysis

and then refinement of a metabolic gene interactome would have been much more impactful.

We thank the reviewer for raising this point. Please note that the rationale of the study was not to discover gene interactions, but to compile well-characterised metabolic pathways as best as we could for metabolic inference. Therefore, we used gene ontology as a starting point and refined it with additional genes based on established data in the literature.

9 genes were added to the glycolysis list: *PDK1*, *PDK3*, *LDHA*, *HIF1A*, *MYC*, *PYGL*, *PGM1*, *LIN28A*, *LIN28B*.

***PDK1* and *PDK3* are kinases, which inhibit the activity of pyruvate dehydrogenase, therefore preventing entry of pyruvate into the Krebs cycle. *LDHA* oxidises pyruvate into lactate and prevents shunt of pyruvate into the Krebs cycle. These genes have been previously used as markers of glycolytic metabolism¹. *PYGL* is an enzyme that catalyses the generation of glucose-1-phosphate from glycogen, which can then feed into the glycolysis pathway. It was profiled before in relation to increased glycolytic metabolism². *LIN28A* and *LIN28B* are important pluripotency factors that inhibit oxidative metabolism, leading to increase in glycolysis. *MYC* was shown to potentiate aerobic glycolysis, potentially by regulation of *LIN28A/B* expression^{3,4}. *PGM1* catalyses conversion of glucose-1-phosphate to glucose-6-phosphate.**

10 genes were added to the OxPhos list: *CTP1A*, *IDH1*, *ACO2*, *UQRC2*, *STAT3*, *EGLN1*, *COX16*, *COX17*, *ATP5G1*, *ATP5G3*).

Two of those are part of the mitochondrial complex IV (*COX16*, *COX17*, cytochrome C oxidase) and another two constitute the ATP synthase complex. *UQRC2* is a subunit of complex III (ubiquinol-cytochrome c reductase). *CTP1A1* is important for oxidative metabolism (specifically oxidation of fatty acids) and was used as a marker for oxidative phosphorylation previously¹. *STAT3* is a crucial regulator of pluripotency in embryonic stem cells and has been shown to regulate mitochondrial metabolism⁵. *ACO2* is the entry enzyme to the TCA cycle that provides NADH₂ and FADH₂ to the electron transport chain⁶. *EGLN1* targets HIF proteins for degradation during normoxia, leading

to increase in oxidative phosphorylation⁷. Finally, *IDH1* is crucial for production of alpha-ketoglutarate, a key TCA metabolite controlling the rate of turnover through the cycle⁸.

In the course of the revisions, we independently validated our approach using a different method for scoring pathway expression (Vision)⁹ and using the Reactome Pathway Database¹⁰, instead of our custom gene set. The metabolic dynamics obtain by these methods are almost identical to the ones shown in the study (Extended Data Fig. S1C).

We also generated gene networks of metabolic pathways sets and present the changes in glycolysis and OxPhos expression between E3.5 and E6.5 in the mouse embryo (Extended Data Fig. 1a, b).

The data presented are well described and figures are generated very nicely. They are clear and support the overall hypothesis, although given the wealth of data describing embryo metabolism during the preimplantation stage, this is not surprising. One question that is, in my view, not adequately addressed is regarding the expression changes that accompany “a conserved metabolic transition at the time of implantation” – this is a difficult point to make from the current set of data (notably with the inclusion of the pig) – indeedm the data doesn’t support the proposition that this conserved implantation switch occurs, because the pig undergoes a rapid and protracted period of elongation that does not attach until much later than the other species studied. Thus, the metabolic switch (which is interesting) is surely INDEPENDENT of implantation? Consideration of other elongating species (i.e. the bovine) would help further understand this observation. This may be relevant in the context of Figure 3, which shows that porcine TE and hypoblast retains relatively high expression of Oxphos, indicative of divergence in the regulation.

We thank the reviewer for raising this point and agree that the metabolic switch is indeed INDEPENDENT of embryo implantation.

In the original version of the manuscript, we had taken a human and non-human primate-centric point of view, leading us to relate the observed metabolic shift to primate CS5. We have corrected this throughout the revised version of the manuscript and consistently relate the shift from a bivalent towards a glycolytic metabolism now to embryonic disc formation or early gastrulation stages.

We also emphasise this point in the abstract:

“To address this issue, we compiled single-cell embryo profiling data in six mammalian species and determined their metabolic dynamics through glycolysis and OxPhos associated gene expression. Strikingly, we identify a conserved switch from bivalent respiration in the late blastocyst towards a glycolytic metabolism in early gastrulation stages across species, which is independent of embryo implantation. Extraembryonic lineages followed the dynamics of the embryonic lineage, except visceral endoderm.”

In the course of the revisions, we have surveyed several bovine RNA-seq datasets¹¹⁻¹³ but unfortunately found that there was no compatible Smart Seq2 single-cell embryo dataset available for bovine embryos. However, we have discovered and included a single-cell RNAseq embryo dataset of another elongating mammalian species, the gray short-tailed opossum *Monodelphis domestica*¹⁴.

As the reviewer will be aware, marsupials are considered a transition stage between monotremes and mammals¹⁴ and thus make an excellent model for evolutionary developmental biology. The opossum embryo implants at embryonic day 11, which is well after the initiation of gastrulation¹⁵. Remarkably, we found that opossum embryogenesis follows the same metabolic dynamics as our previously analysed species, including the pig (Fig. 2J,K). This corroborates the result that the switch from bivalent respiration to a glycolytic metabolism occurs at the embryonic disc stage across mammalian species, independently of implantation.

To make this central message readily accessible to the reader, we have added a summary figure (Fig. 5) including all the embryonic stages analysed, the six species and the timings of implantation.

Furthermore, the authors refer to the increase in glycolysis as being a necessary boost for biosynthetic capacity; a hypothesis that is quote established (i.e. the work of Matthew Vander Heiden, who is well cited in this piece), however the authors could surely have actually answered this question within this study by looking as well as the key biosynthetic genes, such as those involved with the Pentose Phosphate Pathway (at a minimum).

We thank the reviewers for this suggestion. We analysed expression of genes associated with the pentose phosphate pathway and found a slight tendency towards an increase in later stages in some species (most pronounced in opossum, pig and human), which may indicate an increase caused by a rise in glycolytic flux (Extended Data Fig. 5a).

This data is now included in the revised manuscript:

“Opossum and pig datasets (Fig. 2H, J) spanned from zygote to early embryonic disc stages (CS5 equivalent) and we calculated OxPhos and glycolysis module scores (Fig 2I, K). Strikingly, OxPhos peaked at the blastocyst stage (CS3) and glycolysis gradually increased from the blastocyst until the embryonic disc stage (CS5), similar to human, monkey and mouse (Fig. 2I, K). We also examined pentose phosphate pathway associated transcripts in all samples and found a modest increase at the later gastrula stages for the majority of species analysed (Fig. S6A).

Collectively, these data suggest that the shift from bivalent to respiratory metabolism in early rodent development is driven by an evolutionary conserved programme, rather than embryo implantation.”

In addition, we profiled several other pathways associated with an increase in glycolytic metabolism, namely amino acid and fatty acid biosynthesis, including glycine, serine and cholesterol synthesis. We found that none of

these were significantly upregulated in mouse embryogenesis at any developmental stage (see reviewer figure below).

Reviewer figure 1: Module scores in mouse for the metabolic pathways indicated.

The authors also describe that physical measurements of metabolites during preimplantation development requires multiple whole embryos: This statement is incorrect. A number of authors from a range of laboratories have described metabolic function of single mammalian embryos (e.g. Gardner, Sakkas, Leese, Sturmey, Krisher, Prather, Thompson, Houghton to name but a few). Indeed, the

authors follow this statement by implying that metabolic inference is needed to study metabolism of embryonic and extraembryonic (i.e. TE) lineages, however this again has been done directly on at least two occasions (i.e. Gopichandran and Leese 2003 and Houghton 2006).

We acknowledge that this statement may have been an oversimplification. We agree that measurements of metabolites in single embryos, including measurements of extraembryonic tissues such as TE at CS3¹⁶⁻¹⁹ are achievable, however, these techniques come with their own set of limitations, e.g. different formulas for culture media used in various labs prevent accurate inter-species comparison of metabolic measurements¹⁹.

Nevertheless, we would maintain that it is currently impossible to perform physical measurements of isolated embryonic or extraembryonic lineages in post-implantation embryos of human and non-human primate species *in situ*. Analysis of metabolism associated genes in extraembryonic tissues is a first step to delineate the metabolic programme in a variety of species.

We have edited the revised version of the manuscript accordingly.

Finally, the authors attempt to utilise data from ESCs to describe the effects of in vitro culture on gene expression. However, in my view, what this section describes is the effect or prolonged in vitro culture on stem cells, which is of interest, but not as a proxy for embryo culture if comparing in vivo vs in vitro conditions. Indeed, a glance at the data sets used indicate that there was some considerable variation between the origins of the embryos, with some being defined as in vivo and some as in vitro derived – including all of the human embryos. This section would benefit from a more nuanced and accurate account of what is being compared, since a real direct comparison of fully invitro derived embryos with closely aligned, stage-matched embryos harvested at contemporaneous time points to understand the true effects of in vitro embryo culture would be of great value – however this is not what has been presented in this work.

We thank the reviewer for highlighting this important point. In the revised version of the manuscript, we have made sure to accurately state and explain the samples that are being compared in Fig. 4. However, we would need to point out that we have supplied a direct comparison of staged matched *in vitro* and *in vivo* datasets in the cynomolgus monkey (Fig. 4I,J,K). In this figure, we compare the Nakamura et al. (2017) dataset²⁰ (in which *in vivo* postimplantation embryos were dissected from the uterus at given time points) with the Ma et al. (2019) dataset²¹ (in which postimplantation embryos underwent prolonged embryo culture *in vitro*). To our knowledge, these two are the only currently existing mammalian datasets that allow such comparison.

To address the reviewers question of whether the observed effects are the result of prolonged *in vitro* culture, we compared glycolysis and OxPhos score in a single-cell human embryonic stem cell dataset at passage 0 and passage 10²² (Extended Data Fig. 5e). We found no significant differences between glycolysis or OxPhos score between passage 0 and passage 10 human embryonic stem cells, indicating that prolonged culture does not seem to have a significant effect on energy metabolism. We feel that this was an important point to clarify and have included this data into the revised version of the manuscript.

Reviewer #1 (Remarks to the Author):

The central claim of the paper is "a conserved switch from bivalent respiration in the late blastocyst towards a predominantly glycolytic metabolism in early gastrulation stages across species" (abstract, p.2, l. 8ff.). The species in question are 5 mammals (mouse, pig, new and old world monkeys, and human) selected for their different (gastrulation-independent) implantation modes and the claim is based on an elaborate and well informed comparison of single-cell transcriptome profiles from equivalent embryonic stages. Additional claims concern comparisons (using datasets from single or selected species) between in vitro and in vivo growth, between embryonic compartments, and between embryonic tissues and different stem cell cultures. The authors are to be commended for their timely effort to draw attention to the marked differences in early development between mammals and for combining their expertise in comparative embryology and bioinformatics. This resulted in a much desired (universal, i.e., Carnegie) stage-related transcriptome atlas for early mammalian development and there is every reason to believe that the bioinformatics part is sound and solid.

We thank the reviewer for their positive assessment of our manuscript.

The correlation, however, of gene expression profiles with gastrulation and implantation could be rendered more convincing by a systematic description and presentation of the materials and the results and by a rigorous logic connection between published results and the study design.

Major points:

1) Apart from overall tissue architecture and modes of implantation, early mammalian development varies substantially with regard to size of the (vigorously growing) embryonic disc (macaque and human are less than half in size and in cell numbers compared to pig, callithrix and mouse at the beginning of gastrulation) and developmental time spent until gastrulation starts (e.g. 6.5 days in mouse, 10 days in pig, and up to 50 days in callithrix). This needs to be taken into account next to the

timing (and topography) of implantation when considering proliferation and mode of ATP generation, for example.

We thank the reviewer for this comment and agree that embryo size and timing are key considerations. Therefore, we provide consistent Carnegie stage-based annotation of embryonic transcriptomes for stage-matched comparisons (Fig. 2). In the revised version of the manuscript, we have included the short gray-tailed opossum (*Monodelphis domestica*) into our study as another example of a late implanting mammal. Primitive streak formation in *Monodelphis* begins around E10 and similarly to the pig, opossums implant at somatic stages¹⁵. We agree that the length of development spent pre-gastrulation might be a contributing factor for energy metabolism. Indeed, the size of the embryo is also an important consideration. However, we do not see a striking correlation between developmental time spent prior to gastrulation and metabolic trends or embryo size. Overall, these results suggest that energy metabolism in embryonic cells is regulated by an intrinsic programme shared by the mammalian species investigated in our study.

We elaborate on this point in the discussion:

“We show that all six mammalian embryos undergo a conserved transcriptional transition of metabolic gene expression from bivalent towards predominantly glycolytic metabolism at the time of embryonic disc formation. This metabolic switch occurs despite an evolutionary distance of 159 million years as well as different mechanisms and time points of embryo implantation. Furthermore, different species initiate gastrulation at different timelines, with primitive streak formation occurring as early as E6.5 in the mouse and as late as E14-16 in the marmoset^{45,56}. Mammalian embryogenesis is therefore characterised by an intrinsic metabolic programme, conserved regardless of embryo size or timing of – or even developmental time spent prior to - gastrulation^{54,58,62,68,69}. “

Please note that marmoset gastrulation commences between E14-16, as described in our related manuscript (e.g. anterior – posterior axis and pronounced *TBXT* and *MIXL1* expression at CS6, Bergmann et al., in revision).

We have also included a new summary figure (Fig. 5) to clearly illustrate the overarching metabolic programme and the individual dynamics of embryonic development and timing of implantation.

2) The introduction should clarify the priority of the many problems addressed or, indeed, to be solved: improving culturing conditions, differences in in vivo/in vitro development, interspecies tissue architecture comparisons, pre- and post-implantation differences, embryonic-extraembryonic lineage differences, equivalence of various types of stem cells with embryonic lineages etc.. Full credit should be given (s. page 3, bottom half) to Krisher and Prather (2012) for their detailed discussion on the metabolic phenotypes of early mammalian embryos (mainly pig and cow, though) and the Warburg effect, while O'Farrel's (2004) needs to be addressed as dealing with mouse and rat, only, (and not with observations "throughout the mammalian kingdom").

We agree and have adjusted the introduction accordingly. The introduction now clearly highlights the questions to be addressed in the study.

Paragraph 1 ends on: “*Thus, the regulation of glycolysis and OxPhos in the pre- to postimplantation transition in vivo has remained poorly understood.*”

Paragraph 3 concludes that “*it is currently unclear how species-related differences, including mode and timing of implantation, affect the metabolism in embryonic and extraembryonic lineages.*”

And we have added a new paragraph to introduce the importance of delineating *in vitro* versus *in vivo* development:

“Recent protocols for human^{31–33} and non-human primate^{34,35} blastocyst culture to postimplantation stages have opened new avenues to study primate embryo implantation in vitro. The challenge is now to identify the most physiological culture conditions for natural as well as stem cell-based^{36–42} embryo models. However, the metabolic state of embryonic and extraembryonic lineages in vitro has remained elusive. “

We have incorporated and discuss the reference for Krisher and Prather²⁶ and confined O’Farrel’s conclusions²⁷ to rodents in the introduction:

“Similar to other rapidly proliferating cells, pluripotent cells in the early mammalian embryo equally face the challenge to balance energy production with biosynthetic demands²⁹. The murine pluripotent epiblast undergoes accelerated proliferation prior to gastrulation^{30,31}, shifting its metabolism from bivalent, utilizing both glycolytic and OxPhos pathways, towards a predominantly glycolytic flux.”

3) In material and methods, species names should be given when referring to datasets (p.3, 3rd paragr.) and Table 1 should be divided into a list of (a) embryos and (b) all (stem) cell cultures that were analysed. It is not clear whether the results section on stem cell lines (p. 10) are derived from published datasets or whether datasets were created for this study. Methodology for HIF1A and LDHA should be given here (and the rationale for the analysis explained in the Introduction). Why was the cynomolgus dataset by Niu et al. (2019) published in the same issue back-to-back with Ma et al. (2019) not included (or mentioned, at least)?

We have updated Supplementary table 1, dividing it into (A) a list of embryo and (B) a list of stem cell culture samples used in our study, including references and species names. In addition, the revised version of the methods section now includes species names and a description for HIF1A and LDHA expression. The rationale for including HIF1A and LDHA was to clearly demonstrate the profound increase in their expression in the postimplantation embryo due to their important role for a glycolytic shift in embryonic disc / gastrula stage embryos.

The Niu et al. (2019) dataset²⁸ was not included in the study because the raw dataset in the repository is poorly annotated. We have written to the authors to clarify their sample annotation, but unfortunately did not receive a response. Nevertheless, we have now included a reference for the Niu et al. 2019 study in the revised version of the manuscript.

4) Stating that "the five species ... adhered to a conserved metabolic transition at the time of implantation" is an oversimplification at least in the case of callithrix (which implants well after the peak at CS3) and pig (which implants even later: during high-somite-number stages).

We thank the reviewer for this important comment and agree that this point was unclear in the previous version of the manuscript. Our data suggest that metabolic trends in mammalian embryogenesis are conserved in mammalian species regardless of the mode or time of implantation. We believe that the metabolic gene expression is largely correlated to embryonic stage (i.e. epiblast vs embryonic disc) rather than implantation time. We have corrected these statements in the main text, e.g. in paragraph 1 of Discussion:

"We show that all the mammalian embryos investigated undergo a conserved metabolic transition from bivalent towards respiratory metabolism at the time of embryonic disc formation. This metabolic switch occurs despite an evolutionary distance of 159 million years as well as different mechanisms and time points of embryo implantation."

As previously mentioned, we have now included the opossum dataset to provide another species where the blastocyst elongates and which implants after gastrulation¹⁵, similarly to the pig. We discovered that the opossum follows the same metabolic dynamics when stage matched to other five species of the previous datasets (Fig. 2). To further clarify this central message, we have generated a summary figure (Fig. 5) for the discussion.

Minor points

1) OxPhos should be used throughout instead of using varying this term so central to the study.

We have corrected this.

2) Fig. 1 does not show a human dataset (p.5, 3rd paragr.).

Thank you for spotting this error – it was meant to be Fig. 2 and we have corrected this.

3) The Results and Discussion section would be easier to understand (and probably transport the evidence more clearly) if stripped of methodological or introductory descriptions (e.g. p.7, 1st two sentences; p.7 last paragr, and p.8 1st paragr).

We agree and have improved the flow of the main text. In consultation with the editor (*Dr. Ann Le Good*), we have generated separate “Results” and “Discussion” sections in the revised version of the manuscript to transport the evidence more clearly. We decided to keep some of the sentences the reviewer has highlighted in the main text to provide a rationale for our analysis (e.g. p.7, 1st two sentences; p.7 last paragraph), but have removed all other elements of discussion or introduction (e.g. p.8 1st paragraph on implantation modes) from the Results section.

3) Bibliographical details are incomplete in high number of references.

These errors have now been corrected.

References

- 1. Tischler, J. *et al.* Metabolic regulation of pluripotency and germ cell fate through α -ketoglutarate. *EMBO J.* 38, (2019).**
- 2. Zhou, W. *et al.* HIF1 α induced switch from bivalent to exclusively glycolytic metabolism during ESC-to-EpiSC/hESC transition. *EMBO J.* 31,**

2103–2116 (2012).

3. Folmes, C. D. L., Dzeja, P. P., Nelson, T. J. & Terzic, A. Metabolic plasticity in stem cell homeostasis and differentiation. *Cell Stem Cell* vol. 11 596–606 (2012).
4. Zhang, J. *et al.* LIN28 Regulates Stem Cell Metabolism and Conversion to Primed Pluripotency. *Cell Stem Cell* 19, 66–80 (2016).
5. Carbognin, E., Betto, R. M., Soriano, M. E., Smith, A. G. & Martello, G. Stat3 promotes mitochondrial transcription and oxidative respiration during maintenance and induction of naive pluripotency. *EMBO J.* 35, 618–634 (2016).
6. Ciccarone, F. *et al.* Aconitase 2 inhibits the proliferation of MCF-7 cells promoting mitochondrial oxidative metabolism and ROS/FoxO1-mediated autophagic response. *Br. J. Cancer* 122, 182–193 (2020).
7. To, K. K. W. & Huang, L. E. Suppression of hypoxia-inducible factor 1 α (HIF-1 α) transcriptional activity by the HIF prolyl hydroxylase EGLN1. *J. Biol. Chem.* 280, 38102–38107 (2005).
8. M. Gagné, L., Boulay, K., Topisirovic, I., Huot, M. É. & Mallette, F. A. Oncogenic Activities of IDH1/2 Mutations: From Epigenetics to Cellular Signaling. *Trends in Cell Biology* vol. 27 738–752 (2017).
9. DeTomaso, D. *et al.* Functional interpretation of single cell similarity maps. *Nat. Commun.* 10, 4376 (2019).
10. Jassal, B. *et al.* The reactome pathway knowledgebase. *Nucleic Acids Res.* 48, D498–D503 (2020).
11. Lavagi, I. *et al.* Single-cell RNA sequencing reveals developmental heterogeneity of blastomeres during major genome activation in bovine embryos. *Sci. Rep.* 8, (2018).
12. Desmet, K. L. J. *et al.* Oocyte maturation under lipotoxic conditions induces carryover transcriptomic and functional alterations during post-hatching development of good-quality blastocysts: Novel insights from a bovine embryo-transfer model. *Hum. Reprod.* 35, 293–307 (2020).

13. Mamo, S. *et al.* RNA sequencing reveals novel gene clusters in bovine conceptuses associated with maternal recognition of pregnancy and implantation. *Biol. Reprod.* 85, 1143–1151 (2011).
14. Mahadevaiah, S. K., Sangrithi, M. N., Hirota, T. & Turner, J. M. A. A single-cell transcriptome atlas of marsupial embryogenesis and X inactivation. *Nature* 586, 612–617 (2020).
15. Mate, K. E., Robinson, E. S., Vandeberg, J. L. & Pedersen, R. A. Timetable of in vivo embryonic development in the grey short-tailed opossum (*Monodelphis domestica*). *Mol. Reprod. Dev.* 39, 365–374 (1994).
16. Gopichandran, N. & Leese, H. J. Metabolic characterization of the bovine blastocyst, inner cell mass, trophectoderm and blastocoel fluid. *Reproduction* 126, 299–308 (2003).
17. Hewitson, L. C. & Leese, H. J. Energy metabolism of the trophectoderm and inner cell mass of the mouse blastocyst. *J. Exp. Zool.* 267, 337–343 (1993).
18. Houghton, F. D. Energy metabolism of the inner cell mass and trophectoderm of the mouse blastocyst. *Differentiation* 74, 11–18 (2006).
19. Thompson, J. G., Brown, H. M. & Sutton-McDowall, M. L. Measuring embryo metabolism to predict embryo quality. *Reprod. Fertil. Dev.* 28, 41 (2016).
20. Nakamura, T. *et al.* Single-cell transcriptome of early embryos and cultured embryonic stem cells of cynomolgus monkeys. *Sci. Data* 4, 170067 (2017).
21. Ma, H. *et al.* In vitro culture of cynomolgus monkey embryos beyond early gastrulation. *Science* 366, (2019).
22. Yan, L. *et al.* Single-cell RNA-Seq profiling of human preimplantation embryos and embryonic stem cells. *Nat. Struct. Mol. Biol.* 20, 1131–1139 (2013).
23. Hassoun, R., Schwartz, P., Feistel, K., Blum, M. & Viebahn, C. Axial differentiation and early gastrulation stages of the pig embryo.

Differentiation 78, 301–311 (2009).

24. Nakamura, T. *et al.* A developmental coordinate of pluripotency among mice, monkeys and humans. *Nature* 537, 57–62 (2016).
25. O’Rahilly, R. & Müller, F. Developmental Stages in Human Embryos: Revised and New Measurements. *Cells Tissues Organs* 192, 73–84 (2010).
26. Krisher, R. L. & Prather, R. S. A role for the Warburg effect in preimplantation embryo development: Metabolic modification to support rapid cell proliferation. *Mol. Reprod. Dev.* 79, 311–320 (2012).
27. O’Farrell, P. H., Stumpff, J. & Tin Su, T. Embryonic Cleavage Cycles: How Is a Mouse Like a Fly? *Curr. Biol.* (2004) doi:10.1016/j.cub.2003.12.022.
28. Niu, Y. *et al.* Dissecting primate early post-implantation development using long-term in vitro embryo culture. *Science* (80-.). 366, (2019).

Reviewers' Comments:

Reviewer #1:

Remarks to the Author:

As indicated in my first review, this is an interesting analysis of existing data sets and has highlighted some fascinating parallels across mammalian species in terms of expression profiles of genes related to metabolic activity. These data largely confirm a wealth of existing phenotypic measurements.

The authors have made a fine job of addressing all of my comments and queries from the first round of review and the manuscript is now much improved.

Reviewer #2:

Remarks to the Author:

All concerns of this reviewer are addressed appropriately in the revised version.